# Mechanisms Underlying the Development of Murine T-Cell Lymphoblastic Lymphoma/Leukemia Induced by Total-Body Irradiation

**DOI:** 10.3390/cancers16122224

**Published:** 2024-06-14

**Authors:** Toshihiko Sado, John B. Cart, Chang-Lung Lee

**Affiliations:** 1National Institute of Radiological Sciences, Chiba 263-0024, Japan; 2Department of Radiation Oncology, Duke University School of Medicine, Durham, NC 27710, USA; 3Department of Pathology, Duke University School of Medicine, Durham, NC 27710, USA

**Keywords:** radiation, thymic lymphoma, T-cell lymphoblastic lymphoma, T-cell lymphoblastic leukemia, Notch, Ikzf1, Pten, c-Myc

## Abstract

**Simple Summary:**

Exposure to ionizing radiation can increase the excess risk of developing hematologic malignancies. Thymic lymphoma induced by fractionated total-body irradiation is one of the most robust models to study the biology of radiation-induced blood cancers in mice. In this review article, we will discuss results from published papers that elucidate cell-autonomous and non-cell-autonomous effects of ionizing radiation on the initiation, progression, and malignant transformation of T-lineage lymphoma/leukemia in the mouse thymus.

**Abstract:**

Exposure to ionizing radiation is associated with an increased risk of hematologic malignancies in myeloid and lymphoid lineages in humans and experimental mice. Given that substantial evidence links radiation exposure with the risk of hematologic malignancies, it is imperative to deeply understand the mechanisms underlying cellular and molecular changes during the latency period between radiation exposure and the emergence of fully transformed malignant cells. One experimental model widely used in the field of radiation and cancer biology to study hematologic malignancies induced by radiation exposure is mouse models of radiation-induced thymic lymphoma. Murine radiation-induced thymic lymphoma is primarily driven by aberrant activation of Notch signaling, which occurs frequently in human precursor T-cell lymphoblastic lymphoma (T-LBL) and T-cell lymphoblastic leukemia (T-ALL). Here, we summarize the literature elucidating cell-autonomous and non-cell-autonomous mechanisms underlying cancer initiation, progression, and malignant transformation in the thymus following total-body irradiation (TBI) in mice.

## 1. Introduction

Epidemiologic studies reveal a significant association between exposure to ionizing radiation and increased risk of hematologic malignancies in myeloid and lymphoid lineages [1]. Radiation-related excess risk of blood cancers is influenced by multiple factors, including radiation dose, dose rate, radiation volume, biological sex, and the age of exposure [2,3]. For example, the treatment of certain solid tumors with radiation therapy has a strong association with acute myeloid leukemia and myelodysplastic syndromes [4,5]. In addition, atomic bomb survivors have an excess risk of developing various types of blood cancers including acute myeloid leukemia (AML), myelodysplastic syndrome (MDS), and acute lymphoblastic leukemia (ALL) [6,7,8]. One remarkable finding from atomic bomb survivor studies is that the relative risk of blood cancers is approximately 70 times higher among children exposed at the age of 10 or younger, and rapidly decreased in individuals exposed at older ages [6,9]. Sasaki reported comparable murine data on the age dependency of radiation-induced malignant lymphoma, myeloid leukemia, and solid cancers [10]. Given that substantial evidence links radiation exposure with the risk of hematologic malignancies, it is imperative to deeply understand the mechanisms underlying radiation-induced carcinogenesis of the hematopoietic system, especially for individuals exposed to radiation in childhood.

Experimentally, exposure of 4-week-old C57 black mice (equivalent to ~12-year-old humans) to fractionated total-body X- or γ-irradiation (FX) induces lymphomas arising from immature thymocytes [11,12,13,14,15,16,17]. This model was first described by Dr. Henry Kaplan and colleagues in the 1950s, and their findings have been reproduced in many laboratories since then. For example, the results from our laboratories show that exposure of 4-week-old C57BL/6 mice to a single fraction of 1.8 Gy TBI per week for 4 consecutive weeks (1.8 Gy × 4) causes more than 90% of mice to develop thymic lymphomas within about 8 months after irradiation [18,19,20,21]. Lymphoma cells develop primarily in the thymus and express T-cell markers such as CD3, CD4, and/or CD8. In a subset of mice, lymphoma cells become leukemic and disseminate to other organs including the liver, the spleen, the kidney, and/or the bone marrow (BM) [18,20]. The sensitivity of mice to radiation-induced thymic lymphomas varies significantly among different mouse strains. C57BL/6 (B6) and its closely related B10 strain [11,12,13,14,15,16,17,22,23,24,25,26,27,28,29,30,31] as well as BALB/c [32], RFM [33], and Swiss Albino [34] strains are known to be susceptible to FX-induced thymic lymphomas, whereas C3H [35,36], STS [36,37,38,39], and MSM [39,40,41,42] strains are resistant. B6C3F1 mice, which are F1 hybrids of susceptible B6 and resistant C3H strains, are known to be susceptible [43,44]. Because of the robustness of generating radiation-induced blood cancers in wild-type mice, this mouse model of radiation-induced thymic lymphoma has been extensively used to study the mechanisms of radiation-induced carcinogenesis in hematopoietic cells over the past 70 years.

## 2. Genetic Alterations of Murine Thymic Lymphoma Induced by FX Treatment

During the last two decades, much progress has been made in analyzing chromosomal alterations and the mutational landscape of thymic lymphomas induced by FX treatment in various strains of mice. For example, Yoshida et al. reported the results of their extensive analysis of chromosome aberrations detected among FX-induced thymic lymphomas [43]. They found frequent interstitial deletions of chromosome 11, which contains an *Ikaros* tumor suppressor gene, unbalanced translocation with deletion of chromosome 12, which contains the *Bcl11b* tumor suppressor gene, and trisomy or partial trisomy of chromosome 15, which contains the *c-Myc* oncogene. They also noted that chromosomally aberrant cells were present in 25 of 26 (96%) of the thymic lymphomas examined, and the most frequent abnormality observed was trisomy or partial trisomy of chromosome 15 (16/26, 62%), which was consistent with earlier studies [45,46,47,48,49]. In addition, genomic alterations of the *Ikaros*/*IKZF1* tumor suppressor gene and *Notch1* oncogene were frequently observed in murine radiation-induced thymic lymphomas [50,51,52,53,54]. With the use of the distinct patterns of rearrangements at the T-cell receptor β (TCR-β) loci, Ohi et al. found that clonal expansions were detected in a fraction of atrophic thymuses 30 days after irradiation [55]. They further noted a loss of heterozygosity (LOH) at *Bcl11b* and trisomy of *Myc* at high frequencies in both lymphomas and atrophic thymuses. These data suggested that the order of genetic changes that occur during lymphomagenesis was *Bcl11b* and *Myc* at early stages and *Ikaros*, *Pten*, and *Notch1* at later stages.

To examine the mutational landscape of murine radiation-induced thymic lymphomas, we conducted whole-exome sequencing (WES) to determine somatic mutations and copy number alternations in thymic lymphomas developed in wild-type (C57BL/6J) mice, Kras^LA1^ mice containing a somatic *Kras^G12D^* mutation, and p53^+/−^ mice that lose one copy of the tumor suppressor p53 in all somatic cells [20]. Paired normal tissues from the same mice were included as a control for the differences in germline variants. The code for replicating the figures and statistical analyses of WES results in this study are available through a public source code repository (https://gitlab.oit.duke.edu/dcibioinformatics/pubs/kirsch-lee_lymphoma, accessed on 20 April 2020). Our findings from WES demonstrate that the frequency of genetic alterations in the Notch signaling pathway is influenced by the presence of functional p53 [20]. In lymphomas that retained wild-type (WT) p53, approximately 83% of these tumors harbored mutations in *Notch1* and/or *Ikzf1*, a negative regulator of Notch1 signaling. In contrast to p53 WT lymphomas, mutations in *Notch1* or *Ikzf1* only occurred in around 38% of p53 deficient lymphomas. Nonsynonymous mutations in *Notch1* were exclusively found in exon 27, which encodes the heterodimerization domain or HD (all missense mutations), and exon 34, which encodes the proline, glutamic acid, serine, threonine-rich, or PEST domain (missense, stop-gain, and/or frameshift mutations). Also, nonsynonymous mutations in *Ikzf1* were exclusively found in exons 5/6 that encode the DNA binding domain (all missense mutations) and exon 9 (stop-gain or frameshift mutations). These mutations recapitulate hotspot mutations in human *NOTCH1* and *IKZF1* observed in acute lymphoblastic leukemia patients [56,57]. Collectively, genetic analyses of murine radiation-induced thymic lymphomas reveal a high frequency of somatic mutations and copy number variations in critical regulators of Notch signaling, which also occurs frequently in human T-LBL and T-ALL.

## 3. DNA Methylation of FX-Induced Thymic Lymphomas in Mice

In addition to genomic alterations, the development of radiation-induced thymic lymphoma can also be influenced by DNA methylation. For example, exposure of 45-day-old C57BL/6 mice to either acute (5 Gy) or fractionated (0.5 Gy/day × 10 days) TBI caused a significant reduction in global cytosine DNA methylation 6 h after irradiation. Radiation-induced DNA hypomethylation in the thymus persisted up to one month after acute exposure in both males and females [58]. These findings suggest that epigenetic changes in the genome may occur during thymic lymphomagenesis following FX treatment [58]. Indeed, Herranz et al. studied the effect of administration of Zebularine (1-[beta-D-ribofuranosil]-1,2-dihydropyrimidin-2-1), a DNA methylation inhibitor, on the development of thymic lymphomas [59]. They found that treatment with zebularine led to a decrease in 5-methylcytosine genomic content, demethylation of the hypermethylated CpG islands of tumor suppressor genes *RASSF1A* and *p15^INK4b^*, and a significant extension of tumor-free survival. DNA hypomethylation induced by zebularine occurred in association with depletion in extractable DNA methyltransferase 1 protein. They also observed that zebularine did not cause notable toxicity in nonirradiated control mice. Together, these data indicate that hypermethylation of certain tumor suppressor gene(s) is involved during the development of FX-irradiation-induced thymic lymphomas, although the underlying mechanisms remain incompletely understood.

Malumbres et al. reported that inactivation of the cyclin-dependent kinase inhibitor *p15^INK4b^* due to LOH, as well as de novo methylation with independence of *p16^INK4a^* alterations, was observed among murine radiation-induced thymic lymphomas [60,61]. Some years later, Santos et al. also observed that *Cd95* and *Pten*, two genes mapped at the region in chromosome 19 and known to contain thymic lymphoma suppressors (Thymic Lymphoma Suppressor Region 8, TLSR8), were inactivated in a vast majority of these tumors (85.3% for *Cd95* and 61.8% for *Pten*) [62]. These findings and the lack of mutations in the coding sequences of these genes suggest a possible regional epigenetic inactivation mechanism on mouse chromosome 19 operating during the development of radiation-induced thymic lymphomas. Song et al. also reported clear evidence for the methylation state of the promoter region of the *p16* tumor suppressor gene among thymic lymphomas induced by FX treatment [63]. They identified 23 CpG sites of the CpG islands in the p16 promoter region and found that the methylation percentages of −71, −63, −239, −29, −38, −40, −23, and 46 CpG sites were significantly higher in radiation-induced thymic lymphoma tissue than those in matched non-irradiated thymus tissue samples. They suggested that the methylation of these CpG sites in the *p16* promoter may reduce its expression in the thymic lymphomas induced by FX. On the other hand, Yamaguchi et al. reported no evidence for methylation-associated silencing of *Pten*. Instead, their results showed complex structural abnormalities comprising missense and nonsense mutations, 1 and 3 bp insertions, and focal deletions in 8 of 23 lymphomas (35%) [64]. It is important to note that in the experiments conducted by Song et al., the thymic lymphoma-induction-susceptible male BALB/c strain was used [63], while Yamaguchi et al. used female B6C3F1 mice, which are F1 hybrids of TL-induction-susceptible B6 and resistant C3H strains [64]. Therefore, these results suggest that genetic and epigenetic modifications of tumor suppressor genes can cooperate during the development of thymic lymphomas induced by FX treatment. These observations also reveal the potential impact of mouse strains on the mechanisms of gene silencing during radiation-induced thymic lymphomagenesis.

## 4. Dynamic Changes in the Production of Reactive Oxygen Species in the Thymus Following FX Treatment

In 2013, Tsuji et al. published the results that examine the dynamic changes in the cell population that occurred within the thymuses of B6 female mice at various time points after FX treatment [52]. Their analysis included the course of the dynamic cytogenetic as well as genomic changes in the thymocyte populations, and clonal analysis of pre-lymphoma and established lymphoma cells using TCRβ gene rearrangements. In this study, they noted that reactive oxygen species (ROS) were generated in descendants of irradiated thymocytes during recovery from radiation-induced thymic atrophy. Concomitantly, these regenerating thymus cells exhibited lesions of genomic alterations revealed by the appearance of γ-H2AX foci, chromosomal instability, and aneuploidy with trisomy 15. They also observed bystander effects on the induction of chromosome aberrations in co-cultured ROS-sensitive *XRCC4*^−/−^, *OGG1*^−/−^ and *Mth1*^−/−^ mutant cells. The disappearance of these bystander effects by superoxide dismutase and catalase suggested the role of ROS generated from post-irradiation thymocytes. Trisomy 15 and aberrant karyotypes were also observed in high frequency among these thymus cells. The emergence of thymic lymphomas from the thymocyte population containing abnormal cell clones supports that clones with trisomy 15 and altered karyotypes were pre-lymphoma cells with the potential to develop into thymic lymphomas. It was noted that alterations of the oncogene *Notch1* were observed after the pre-lymphoma cells were established. In addition, Kominami and colleagues reported that *Mtf-1* (metal-responsive transcription factor-1) allelic genes, which are involved in the regulation of the response of cells to heavy metals and cellular ROS, play an important role in regulating the sensitivity of mice to radiation-induced thymic lymphomas [39,40,41,42]. Their results showed that a high level of ROS was observed in large thymocytes that survived exposure to radiation in the thymic lymphoma-susceptible BALB/c strain, but not in thymocytes of the thymic lymphoma-resistant MSM strain. Notably, one study by Zhao et al. described the protection effect of hydrogen on radiation-induced thymic lymphoma in BALB/c mice. They observed that pretreatment of mice with hydrogen before TBI decreased ROS levels induced by irradiation [65]. Together, these findings reveal an important contribution of chronic ROS to the development of thymic lymphoma following FX treatment.

## 5. Detection of Pre-Lymphoma Cells within the Thymus of FX-Treated Mice

To dissect the cellular mechanisms of thymic lymphomagenesis in FX-treated B10 mice, Sado and colleagues attempted to distinguish between “pre-lymphoma cells” and established “thymic lymphoma cells” with the use of intrathymic (i.t.) and intraperitoneal (i.p.) injection of cells isolated from the thymus at varying time intervals after FX treatment. They found that thymic lymphoma cells from mice 3 months after FX treatment into sub-lethally irradiated (4 Gy) Thy 1 congenic sex-matched recipient mice via i.t. and i.p. injections grew vigorously in both sites. In contrast, when thymus cells harvested 3 weeks after FX treatment were used, they only grew when injected intrathymically. These results indicate that thymus cells of this stage contained cells that were already committed to becoming neoplastic cells but still require the thymic environment to further develop into “autonomous” thymic lymphoma cells. Thus, they termed these cells as “pre-lymphoma cells (PLCs)” [29]. CD markers manifested by these PLCs were CD4^−^CD8^−^, CD4^−^CD8^+^, or CD4^+^CD8^−^ [31]. Using this assay system, they found that as many as 23% of B10 mice contained PLCs in their thymuses when examined 2 weeks after FX treatment. Between 3 and 4 weeks after FX treatment, the frequency of thymuses containing PLCs reached as high as 63%. On the other hand, i.t. injection of BM cells from FX-treated mice within one month after FX treatment never developed donor-derived thymic lymphomas, indicating that the BM is not the initial site where PLCs develop.

## 6. FX Treatment Induces Thymic Lymphoma in Mice through Non-Cell-Autonomous Mechanisms

The observations made by Kaplan and colleagues that lymphomas develop in unirradiated thymic grafts transplanted into a thymectomized irradiated host demonstrate that exposure of the thymus to radiation is not necessary for lymphoma formation. This experimental model has several particularly intriguing features. First, the development of thymic lymphomas induced by FX treatment in mice could be prevented when the BM or spleen was shielded by lead while the rest of the body was irradiated [16,17], or when BM cells from unirradiated mice were injected intravenously shortly after FX treatment [12]. In addition, the development of FX-induced thymic lymphomas could be prevented by thymectomy before or shortly after FX treatment. These findings suggest that although the thymus is where lymphomas develop following FX, the initiation and/or progression of lymphoma cells in the thymus can be suppressed by unirradiated cells derived from the BM. Second, subcutaneous grafting of unirradiated, histocompatible thymuses from neonatal mice into FX-treated, thymectomized mice resulted in the development of thymic lymphomas, most of which were unirradiated thymus-graft-derived [13,14,15]. The authors noted no significant difference in the incidence of lymphoma in thymic grafts implanted 1 h, 1 day, and 8 days after irradiation [66]. The incidence of lymphoma in unirradiated thymic grafts is approximately 40% [14], which is lower than the incidence of thymic lymphoma in situ in C57 black mice irradiated with the same radiation protocol [11]. However, the lymphomas in unirradiated thymic grafts and thymic lymphomas in situ have similar latent periods, morphological appearance, and organ distribution [14]. This observation, often described as an ‘indirect’ induction of radiogenic lymphomas, was considered to suggest an involvement of ‘infectious’ virus-like agents that were released from the host cells by radiation exposure, possibly due to the activation of latent endogenous oncogenic viruses [17,22,23,24,25]. However, attempts to reveal such agents have not been successful [26], as has been reviewed by Sado et al. [28] and Kamisaku et al. [35].

During the late 1970s, Sado and his collaborators at the National Institute of Radiological Sciences (NIRS) in Japan initiated their attempts to unravel the mechanism of FX-induced thymic lymphomas. They utilized B10.Thy 1 congenic pairs of mice that enabled transplantation of the thymus as a whole organ, free thymocytes, or BM cells, from normal or FX-treated donors, into recipients that were variously treated depending on the purpose of each experiment, and then followed the fate of donor- or host-derived cells using Thy 1 alloantigen markers, Thy 1.1 or Thy 1.2. [27,28,29,30,31]. The B10.Thy 1.1 congenic strain of mice used by Sado and his collaborators was bred in their laboratories before they started their thymus grafting and BM transplantation experiments [27] and, therefore, they are original strain combinations initiated by this group. In the first experiment of this series, Muto et al. transplanted thymuses from unirradiated 7-day-old donors (Thy 1.1) into thymectomized recipients (Thy 1.2) that were pretreated with FX-irradiation, and the developed tumors were typed by the use of Thy 1 alloantigen markers [27]. To assess the origin of tumor-initiating cells, 28 out of these 37 lymphomas were typed individually, and they found that 21 out of 28 thymic lymphomas (75%) originated from thymocytes of the unirradiated thymus grafts, whereas 5 tumors (17.5%) were derived from cells of the irradiated hosts. Remarkably, lymphoma cells originating from unirradiated thymus grafts exhibited extensive chromosome abnormalities, which were mainly numerical changes of many chromosomes and polyploidizations, even though they were not exposed to radiation at all.

## 7. Suppressing the Development of Thymic Lymphomas in FX-Treated Mice by Transplantation of Bone Marrow Cells from Unirradiated Donors

Although radiation induces the development of T-cell lymphomas in the thymus, the critical cellular target of radiation is within the BM. Sado et al. analyzed the role of BM transplantation (BMT) on the suppression of thymic lymphoma development in FX-treated mice [28]. All BMT experiments described below were conducted using Thy 1 congenic B10 male mice. In the first experiment of this series, they injected varying numbers (2 × 10^4^~8 × 10^7^) of BM cells from unirradiated B10.Thy 1.1 donors into FX-treated B10.Thy 1.2 mice one day after the last dose of FX treatment. They then examined the tumor incidence/mortality and performed Thy 1 typing of the developed tumors. Their results indicate that the incidence of thymic lymphomas decreased exponentially as the number of BM cells injected was increased. The maximum suppression was observed at the highest BM cell dose employed, i.e., 8 × 10^7^ cells. Results of the Thy 1 typing of the developed tumors in this series indicated that 100% (34/34) were host-derived. This observation indicates that thymocytes derived from the normal BM never become transformed by the putative virus-like agent that has been claimed to be released from the irradiated hosts and causes the transformation of the regenerating immature thymus cells [17,22,23,24,25]. They also showed that when 10^7^ BM cells from normal Thy 1 congenic donor mice were injected immediately after the last dose of FX treatment, the development of thymic lymphomas was not observed, i.e., the development of thymic lymphoma cells was completely suppressed.

Notably, when the time of BMT was delayed for 10 days, the suppressive effect of BMT on lymphoma development was reduced to as low as 30%. Delaying the time of BMT for one month resulted in practically no prevention of the development of thymic lymphomas. This phenomenon is associated with the observation that delaying the time of BMT for 10 days caused a 10-day delay in the repopulation of donor BM-derived thymocytes within the thymuses of the FX-treated, bone-marrow-reconstituted mice. Muto et al. found that the frequency of BM-derived lymphoid progenitors of the FX-treated mice was reduced to as low as 1~2% of the normal BM when examined one month after FX treatment [29]. It has been shown that the suppression of lymphoma development by BMT is primarily due to the inhibition of PLC formation [28]. These results indicate that the formation of PLCs, which normally appear within the thymus between one and two weeks after FX treatment, can be prevented only when the repopulation of the donor-derived thymocytes precedes the appearance of PLCs in the regenerating thymus [28].

As the final experiment of this series, they injected BM cells harvested from B10.Thy 1.1 donor mice treated with FX one month earlier, into lethally (9 Gy) irradiated B10.Thy 1.2 mice. They observed a high incidence of thymic lymphomas in the recipients of irradiated BM, whereas no lymphomas were recovered from the recipients of control unirradiated BM. Importantly, the large majority, as high as ~78%, of the lymphomas developed in the mice that received irradiated BM cells were host-derived (Thy 1.2). Collectively, all these results point to a notion that the role of the BM of FX-treated mice in the development of thymic lymphomas is to create a condition that forces compensatory proliferation of immature T-cell precursors of the host thymus due to a greatly reduced recruitment of BM-derived hematopoietic progenitors induced by FX treatment. This regenerative response to the shortage of BM-derived hematopoietic progenitors in FX-treated mice leads to preneoplastic transformation of immature T-cell precursors of the host thymus through mechanisms that remain incompletely understood.

To better understand the mechanisms by which BM-derived hematopoietic progenitors suppress the development of radiation-induced thymic lymphomas, we conducted transplantation experiments using BM cells from Rag2^−/−^; γc^−/−^ mice, which cannot occupy thymic niches beyond double-negative 2 (DN2) or Rag2^−/−^ mice, which cannot occupy thymic niches beyond DN3 [19]. Our results showed that transplantation with wild-type BM cells significantly protected mice from radiation-induced thymic lymphoma. However, BM cells from either Rag2^−/−^; γc^−/−^ mice or Rag2^−/−^ mice failed to prevent lymphoma development induced by FX treatment. In addition, Rag2^−/−^ BM cells had significant defects in repopulating thymocytes beyond DN3 compared to WT BM. It has been shown that the development of T-cell leukemia is a consequence among X-linked severe combined immunodeficiency (SCID-X1) patients treated with retrovirus-mediated gene transfer [67,68]. Ginn et al. hypothesized that the reconstitution of SCID-X1 patients with limiting numbers of hematopoietic progenitors might be a risk factor for lymphoid malignancy [69]. To test this hypothesis, SCID-X1 mice were reconstituted with serially fewer wild-type hematopoietic progenitors, in the absence of transduction. They found a robust inverse correlation between hematopoietic progenitor cell dose and T-lymphoid malignancy, with earlier disease onset at lower cell dose. Malignancies were of donor origin and carried activating *Notch 1* mutations. They consider that thymocyte self-renewal induced by progenitor deprivation carries an oncogenic risk modulated by intrathymic competition from differentiation-committed cells.

Taken together, the results of several independent laboratories reveal that the BM is not the site of origin of PLCs in FX-treated mice. However, repeated (or fractionated) irradiation of the BM creates a condition that promotes the preneoplastic transformation of regenerating immature thymocytes within the irradiated thymus due to impaired niche competition from irradiated BM-derived hematopoietic progenitors.

## 8. Contributions of Thymus Stromal Cells to Tissue Regeneration in Response to TBI

Five years after Sado’s paper on BMT experiments using Thy 1 congenic donor–host combinations was published, Potworowski et al. reported that when immortalized thymic dendritic cells were injected into FX-treated C57BL/Ka mice, development of thymic lymphomas was significantly suppressed, but not as much as that observed after transplantation of normal BM cells [70]. Based on this observation, they suggested that lymphoma abrogation by BM cells involved the participation of marrow-derived dendritic cells. In this connection, it is important to note that although earlier work suggested an early separation of lymphoids from myeloid lineages during hematopoiesis and hypothesized that the thymus was settled exclusively by lymphoid-restricted hematopoietic progenitors, more recent data have established that lymphoid-myeloid progenitors, which possess both lymphoid and myeloid lineage potentials but lack erythroid potential, are present at the clonal level in early thymic progenitors, indicating that T-cell progenitors settling the thymus are lymphoid–myeloid progenitors [71,72,73,74,75,76]. These findings suggest that the term pre-T cells, used in this paper, are lymphoid–myeloid progenitors that give rise not only to thymic lymphocytes but also to dendritic cells [77] that may partly support the regeneration of thymocytes after TBI.

Defresne et al. reported that when mice were exposed to leukemogenic fractionated doses of irradiation, thymic nurse cells (TNCs), which are specialized epithelial cells that reside in the thymic cortex and are known to have an important role in the generation of thymic lymphocytes, had disappeared [78,79], and they correlated this with the loss of an epithelial cell surface antigen [80,81]. In addition, they observed that epithelial cells have lost their capacity to interact with fetal thymocytes in vitro. Marrow grafting early after irradiation, which prevents the development of lymphomas, restores thymic nurse cells and thymocyte repopulation. Such reconstitution and lymphoma prevention were not observed when marrow grafting was performed 1 month after irradiation. They further noted that, in FX-treated mice, thymic lymphopoiesis was impaired; thymocyte numbers were subnormal, and thymic nurse cells disappeared in a progressive but irreversible fashion. Thus, they considered that depletion of these lymphoepithelial complexes, which are normally involved in the early steps of thymic lymphopoiesis, was related to altered prothymocyte activity in the BM and a damaged thymic environment [80,81]. The grafting of normal BM cells after irradiation prevented the development of lymphomas. At the same time, thymic lymphopoiesis was restored; thymocytes and nurse cell numbers returned to normal because of the proliferation of grafted marrow cells within the thymus. Together, these results suggest an important role of TNC in regulating the regeneration of thymocytes and the development of thymic lymphoma following TBI.

## 9. Possible Cellular Mechanism of Thymic Lymphomagenesis Induced by FX Treatment

Based on the findings summarized in this review article, we propose that the primary cause for the development of thymic lymphomas induced by FX treatment is a shortage in the supply of T-cell precursors (pre-T cells) from the BM to the damaged thymus. Under this condition, the progeny of the residual intrathymic T-cell precursors (pro-T cells) that have survived repeated radiation exposure likely undergo differentiation or maturation arrest [82,83,84]. These pre-malignant cells eventually progress to fully malignant lymphoma cells under the influence of the thymic microenvironment (Figure 1). This model can also explain the ‘indirect’ induction of lymphomas in unirradiated newborn thymuses grafted into FX-treated mice without introducing ‘infectious’ virus-like agents. It is known that when normal newborn thymuses are grafted into histocompatible hosts, most thymus cells present at the time of the grafting undergo disintegration through apoptosis, and residual intrathymic pro-T cells regenerate first and are then replaced with host-derived pre-T cells [85]. Thus, when normal thymuses are grafted subcutaneously into FX-treated recipient mice, which are deficient in pre-T cells in the BM, the regenerating graft-derived pro-T cells must also undergo differentiation arrest, which was followed by preneoplastic transformation.

When the normal cellular differentiation pathway is disrupted acutely post-TBI due to a deficit or reduction in the recruitment of T-cell precursors from the BM, immature thymocytes differentiating into the next stage could be arrested or frozen at a specific stage of the differentiation pathway. The differentiation arrest [78,79,80] or maturation block [86] of those immature thymocytes could be associated with the activation of aberrant expression of ‘emergency’ genes that contribute to lymphoma formation. A similar phenomenon was described by Martin et al. showing that the replacement of thymus-resident precursors with bone-marrow-derived progenitors is regulated by natural cell competition between ‘young’ bone-marrow-derived and ‘old’ thymus-resident progenitors [87]. The authors described that thymus autonomy, a phenomenon where the thymus can produce T cells in the absence of “young” BM-derived lymphoid progenitors, links to a high risk of malignant transformation. It has been shown that thymus autonomy is permissive for preneoplastic transformation [87,88,89]. Another paper from the same group showed that thymus autonomy relies on double-negative 3 early (DN3e) thymocytes that acquire stem-cell-like properties and preserve the hallmarks of thymopoiesis at the single-cell level. Notably, the authors observed a new cell population emerged in thymus autonomy, which probably serves as tumor-initiating cells because these cells express an aberrant Notch1 target gene signature [88].

Several papers published by Boniver and his collaborators show the prevention of thymic lymphomas of FX-treated C57BL mice by injection with tumor necrosis factor-α (TNF-α) after irradiation [90,91,92]. These authors started from the hypothesis that any agents able to restore the thymus after leukemogenic irradiation would exert the same effects as a BM graft. They studied TNF-α as one such possible agent because it was known to modulate the functions of the thymic epithelium and thymocyte subpopulation. Their results indicated that multiple injections of TNF-α restored thymocyte subpopulations and thymic epithelium after irradiation. These changes were associated with a decrease in PLCs and the development of radiation-induced thymic lymphomas. Moreover, BM grafts significantly stimulated intrathymic production of TNF-α messenger RNA, and anti-TNF-α antibodies partially inhibited the anti-lymphoma effects of BM transplantation on FX-treated mice. However, our group reported that BM cells from genetically engineered mice that have impaired tumor immunosurveillance, by deleting TNF-α, interferon γ, or perforin-1, remained sufficient to suppress the formation of radiation-induced thymic lymphoma [19]. Together, these findings suggest that TNF-α treatment and transplantation of unirradiated BM cells may function through independent mechanisms to suppress the development of radiation-induced thymic lymphoma in mice.

A growing body of literature demonstrates the crucial role of p53-mediated cell death in promoting the development of radiation-induced thymic lymphoma. It has been shown that radiation-induced thymic lymphoma can be prevented through various loss-of-function approaches that block the activation of p53-mediated cell death in response to acute DNA damage, including deletion of transcriptional targets of p53 including PUMA [93,94] and Irf5 [95], mutation of Mdm2 [96], and temporary knockdown of p53 during FX treatment [18]. One common observation across these mouse models is that inhibition of p53-mediated apoptosis protects BM-derived hematopoietic progenitors from radiation injury. Therefore, these findings demonstrate that p53-mediated cell death in BM-derived hematopoietic progenitors following FX treatment is essential to induce thymic lymphoma development in mice.

## 10. Conclusions

Collectively, more than 70 years of research on mouse models of radiation-induced thymic lymphomas reveal cell-autonomous and non-cell-autonomous mechanisms underlying cancer initiation and progression. This knowledge provides a strong foundation that we can build upon to deeply understand cellular and molecular changes during the latency period between radiation exposure and the emergence of fully transformed cancer cells. The findings summarized in this review article may also explain the peculiar nature of the radiation dose–response curves reported earlier for BALB/c [32] and C57BL/6 [97] strains, showing that the lowest total dose of TBI for the induction of thymic lymphoma in mice is around 2 Gy. This threshold dose may implicate that this level of radiation damage is required to reduce the number of pre-T cells in the BM to promote the oncogenic transformation of the surviving pro-T cells within the thymus. Furthermore, the mutational landscape of murine radiation-induced thymic lymphomas recapitulates genetic alterations in human *NOTCH1* and *IKZF1* frequently observed in acute lymphoblastic leukemia patients [56,57]. These findings suggest the potential use of this autochthonous mouse model to investigate the biology and novel treatment of human T-ALL and T-LBL.

## Figures and Tables

**Figure 1 cancers-16-02224-f001:**
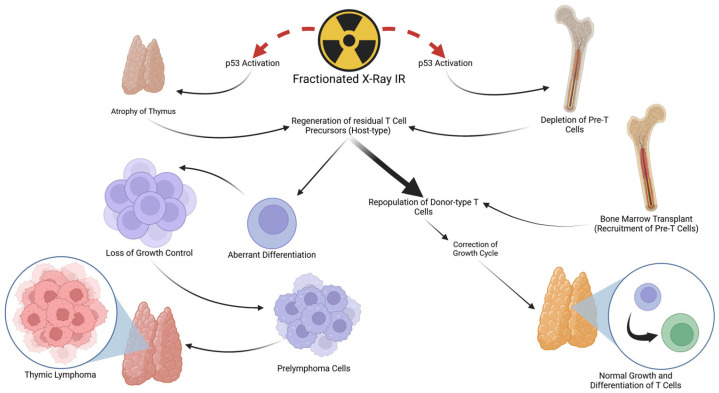
Schematic representation of the possible cellular events that lead to the development of thymic lymphomas after FX treatment. FX treatment induces p53-mediated cell death of BM-derived hematopoietic stem/progenitor cells that impairs the supply of T-cell precursors (pre-T cells) from the BM to the damaged thymus. Under this condition, the progeny of the residual intrathymic T-cell precursors (pro-T cells) that have survived repeated radiation exposure likely undergo differentiation or maturation arrest and thymus autonomy, a phenomenon in which resident thymocytes attempt to sustain T-cell production, but it is also permissive for preneoplastic transformation. These pre-malignant cells eventually progress to fully malignant lymphoma cells under the influence of the altered thymic microenvironment. (Created with BioRender.com).

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
