# Peer review of "Mechanisms Underlying the Development of Murine T-Cell Lymphoblastic Lymphoma/Leukemia Induced by Total-Body Irradiation"

_cancers, 2024, doi:10.3390/cancers16122224_

Round 1
Reviewer 1 Report (Previous Reviewer 2)
Comments and Suggestions for Authors
The revised version is very much improved and authors have incorporated pervious comments to improve the flow, scientific content and clarity. however some comments they still need some attention below are some examples:
1. Some sentences are overly complex, which could be simplified for better readability. For example: The experimental setup, which was meticulously designed to eliminate all potential sources of error, ultimately contributed to the high reliability of the data obtained."; another example, "Given the substantial methodological rigor employed throughout the study, the resultant data is of exceptional reliability," in some instances, provide example: Instead of "The results showed a significant increase," use "The results showed a 25% increase (p < 0.05)."
2. there are instances where supporting evidence is mentioned without sufficient detail. Expanding on these points with specific data and statistical analysis would strengthen the argument. For instance, instead of saying, "The results showed a significant increase," specify how significant the increase was with exact figures and statistical values.
3. Define key terms when they are first introduced, such as "thymus autonomy" and "pre-T cells."
4. Summarize the key findings from referenced studies, particularly when multiple references are cited together
5. Indicate for each of the other mechanisms described what the treatment by TNF-α would do and list some arguments why it will generate a supporting and/or contrasting role of your main hypothesis
Author Response
The revised version is very much improved and authors have incorporated pervious comments to improve the flow, scientific content and clarity. however some comments they still need some attention below are some examples:
We thank Reviewer#1 for stating that “The revised version is very much improved and authors have incorporated pervious comments to improve the flow, scientific content and clarity”. Please see our responses to Reviewer#1’s comments below in blue.
- Some sentences are overly complex, which could be simplified for better readability. For example: The experimental setup, which was meticulously designed to eliminate all potential sources of error, ultimately contributed to the high reliability of the data obtained."; another example, "Given the substantial methodological rigor employed throughout the study, the resultant data is of exceptional reliability," in some instances, provide example: Instead of "The results showed a significant increase," use "The results showed a 25% increase (p < 0.05)."
We wanted to respectfully mention that these sentences Reviewer#1 quoted in this comment are not from our manuscript.
- there are instances where supporting evidence is mentioned without sufficient detail. Expanding on these points with specific data and statistical analysis would strengthen the argument. For instance, instead of saying, "The results showed a significant increase," specify how significant the increase was with exact figures and statistical values.
We wanted to respectfully mention that the sentence Reviewer#1 quoted in this comment is not from our manuscript. In addition, the purpose of this review article is to summarize the key findings from the literature instead of discussing the data in great detail. We included references in which readers could find the actual Figures and Tables.
- Define key terms when they are first introduced, such as "thymus autonomy" and "pre-T cells."
We explained the terms "thymus autonomy" and "pre-T cells" when first described in the manuscript as shown in the sentences below.
From the revised manuscript:
The authors described that thymus autonomy, a phenomenon where the thymus can produce T cells in the absence of “young” BM-derived lymphoid progenitors, links to a high risk for malignant transformation.
These findings suggest that the term pre-T cells, used in this paper, are lymphoid-myeloid progenitors that give rise not only to thymic lymphocytes but also to dendritic cells [79] that may partly support the regeneration of thymocytes after TBI.
- Summarize the key findings from referenced studies, particularly when multiple references are cited together
We thank Reviewer#1 for this comment. We have reviewed our manuscript to ensure that key findings from cited references were clearly described.
- Indicate for each of the other mechanisms described what the treatment by TNF-α would do and list some arguments why it will generate a supporting and/or contrasting role of your main hypothesis
The possible effect of TNF-α was discussed in the sentences below.
From the revised manuscript:
Their results indicated that multiple injections of TNF-α restored thymocyte subpopulations and thymic epithelium after irradiation. These changes were associated with a decrease in PLCs and the development of radiation-induced thymic lymphomas.
Reviewer 2 Report (Previous Reviewer 1)
Comments and Suggestions for Authors
The authors addressed all the concerns.
Author Response
We thank Reviewer#2's time and effort in reviewing our revised manuscript.
This manuscript is a resubmission of an earlier submission. The following is a list of the peer review reports and author responses from that submission.
Round 1
Reviewer 1 Report
Comments and Suggestions for Authors
The interesting review by Toshihiko Sado elucidates the cell-autonomous and non-autonomous effects of ionizing ration, pointing to their role in T-cell lymphoma/leukemia development in murine thymus. They focus on the important role of the thymus and the presence of "pre-lymphoma cells", CD4-CD8-, CD4-CD8+, or CD4+CD8-, and the equilibrium with the bone marrow cell compartment. All relevant molecular events are highlighted.
The paper is well-written, the concepts are clear and well-exposed, and the readability is high.
Only a few concerns as follows:
-page 2 lines 85-89. Consistent with the role of Notch1 mutation, there is any evidence that the genomic alteration can include Notch3? Mutations of this Notch receptor have been involved in T-ALL as indicated in the following papers: Oncogene. 2016 Nov 24;35(47):6077-6086; Genes Chromosomes Cancer. 2017 Feb;56(2):159-167). This could be potentially a new area of study.
-Page 4 paragraph 55. Detection of pre-lymphoma cells within the thymus of FX-treated mice. The interesting data presented here are in line with one paper on the Notch3-induced T-ALL murine model demonstrating that "pre-leukemic cells" escape early from the thymus to colonize bone marrow and spleen. Oncogene. 2018 Dec;37(49):6285-6298.
-Page 6 lines 289-293. The authors point to the role of bone marrow as an important source of T-cell precursors. Could there be a question of imbalance between thymus resident and bone marrow T-cell recruitment?
An interesting paper proposes that cell competition between 'young' bone marrow-derived and 'old' thymus-resident progenitors could be the cause of T-ALL (Nature. 2014 May 22;509(7501):465-70.doi: 10.1038/nature13317.). Indeed, they suggest that disruption of cell competition leads to progenitor self-renewal, upregulation of Hmga1, transformation, and T-cell acute lymphoblastic leukemia (T-ALL) resembling the human disease in pathology, genomic lesions, leukemia-associated transcripts, and activating mutations in Notch1. Can you please briefly comment on it?
-As a general comment I found that no mention is given to the thymic epithelial (TECs) cells in the context of Murine T-cell 2 Lymphoblastic Lymphoma/Leukemia Induced by Total-Body 3 Irradiation, mostly pointing to the thymus niche.
Author Response
Reviewer#1
The interesting review by Toshihiko Sado elucidates the cell-autonomous and non-autonomous effects of ionizing ration, pointing to their role in T-cell lymphoma/leukemia development in murine thymus. They focus on the important role of the thymus and the presence of "pre-lymphoma cells", CD4-CD8-, CD4-CD8+, or CD4+CD8-, and the equilibrium with the bone marrow cell compartment. All relevant molecular events are highlighted.
The paper is well-written, the concepts are clear and well-exposed, and the readability is high.
Only a few concerns as follows:
-page 2 lines 85-89. Consistent with the role of Notch1 mutation, there is any evidence that the genomic alteration can include Notch3? Mutations of this Notch receptor have been involved in T-ALL as indicated in the following papers: Oncogene. 2016 Nov 24;35(47):6077-6086; Genes Chromosomes Cancer. 2017 Feb;56(2):159-167). This could be potentially a new area of study.
-Page 4 paragraph 55. Detection of pre-lymphoma cells within the thymus of FX-treated mice. The interesting data presented here are in line with one paper on the Notch3-induced T-ALL murine model demonstrating that "pre-leukemic cells" escape early from the thymus to colonize bone marrow and spleen. Oncogene. 2018 Dec;37(49):6285-6298.
We thank Reivewer#1 for these suggestions. We examined two independent whole-exome sequencing data we published (PMID: 34035082 and PMID: 37558703). However, we did not observe high-frequency somatic mutations or copy-number alterations in the Notch3 gene in murine radiation-induced thymic lymphomas. Nevertheless, it is possible that Notch3 expression could be altered in these tumors at the transcriptional or translational levels that would warrant future investigations.
-Page 6 lines 289-293. The authors point to the role of bone marrow as an important source of T-cell precursors. Could there be a question of imbalance between thymus resident and bone marrow T-cell recruitment?
An interesting paper proposes that cell competition between 'young' bone marrow-derived and 'old' thymus-resident progenitors could be the cause of T-ALL (Nature. 2014 May 22;509(7501):465-70.doi: 10.1038/nature13317.). Indeed, they suggest that disruption of cell competition leads to progenitor self-renewal, upregulation of Hmga1, transformation, and T-cell acute lymphoblastic leukemia (T-ALL) resembling the human disease in pathology, genomic lesions, leukemia-associated transcripts, and activating mutations in Notch1. Can you please briefly comment on it?
We thank Reviewer#1 for these comments. We have included the following paragraph in the revised manuscript.
“When the normal cellular differentiation pathway is disrupted acutely post-TBI due to a deficit or reduction of the recruitment of T cell precursors from the bone marrow, immature thymocytes differentiating into the next stage could be arrested or frozen at a specific stage of the differentiation pathway. The differentiation arrest78-80 or maturation block81 of those immature thymocytes could be associated with the activation of aberrant expression of ‘emergency“ genes that contribute to lymphoma formation. A similar phenomenon was described by Martin et al showing that the replacement of thymus-resident precursors with bone-marrow-derived progenitors is regulated by natural cell competition between ‘young’ bone-marrow-derived and ‘old’ thymus-resident progenitors82. The authors described that thymus autonomy, a phenomenon where the thymus can produce T cells in the absence of competition from “young” bone marrow-derived lymphoid progenitors, links to a high risk for malignant transformation. Another paper from the same group showed that thymus autonomy relies on double-negative 3 early (DN3e) thymocytes that acquire stem-cell-like properties and preserve the hallmarks of thymopoiesis at the single-cell level. Notably, the authors observed a new cell population emerged in thymus autonomy, which probably serves as tumor-initiating cells because these cells express an aberrant Notch1 target gene signature83.
-As a general comment I found that no mention is given to the thymic epithelial (TECs) cells in the context of Murine T-cell Lymphoblastic Lymphoma/Leukemia Induced by Total-Body Irradiation, mostly pointing to the thymus niche.
We thank Reivewer#1 for bringing this topic up to our discussion. We have included the following paragraphs in the revised manuscript.
“Five years after Sado’s paper on bone marrow transplantation experiments using Thy 1 congenic donor-host combinations was published, Potworowski et al. reported that when immortalized thymic dendritic cells were injected into FX-treated C57BL/Ka mice, development of thymic lymphomas was significantly suppressed but not as much as that observed after transplantation of normal bone marrow cells76. Based on this observation, they suggested that lymphoma abrogation by bone marrow cells involved the participation of marrow-derived dendritic cells. In this connection, it is important to note that although earlier work suggested an early separation of lymphoid from myeloid lineages during hematopoiesis and hypothesized that the thymus was settled exclusively by lymphoid-restricted hematopoietic progenitors, more recent data have established that lymphoid-myeloid progenitors, which possess both lymphoid and myeloid lineage potentials but lack erythroid potential, are present at the clonal level in early thymic progenitors, indicating that T cell progenitors settling the thymus are lymphoid-myeloid progenitors77-82. These findings suggest that the term pre-T cells, used in this paper, are potentially lymphoid-myeloid progenitors that give rise to dendritic cells83 that may partly support the regeneration of thymocytes after TBI.
Defresne et al reported that when mice were exposed to leukemogenic fractionated doses of irradiation, thymic nurse cells (TNCs) which are specialized epithelial cells that reside in the thymic cortex and are known to have an important role in the generation of thymic lymphocytes70,71 had disappeared, and they correlated this with the loss of an epithelial cell surface antigen72,84. In addition, they observed that epithelial cells have lost their capacity to interact with fetal thymocytes in vitro. Marrow grafting early after irradiation, which prevents the development of lymphomas, restores thymic nurse cells and thymocyte repopulation. Such reconstitution and lymphoma prevention were not observed when marrow grafting was performed 1 month after irradiation. They further noted that, in FX-treated mice, thymic lymphopoiesis was impaired; thymocyte numbers were subnormal, and thymic nurse cells disappeared in a progressive but irreversible fashion. Thus, they considered that depletion of these lymphoepithelial complexes, which are normally involved in the early steps of thymic lymphopoiesis, was related to altered prothymocyte activity in bone marrow and damaged thymic environment72,84. The grafting of normal bone marrow cells after irradiation prevented the development of lymphomas. At the same time, thymic lymphopoiesis was restored; thymocytes and nurse cell numbers returned to normal as a consequence of the proliferation of grafted marrow cells within the thymus. Together, these results suggest an important role of TNC in regulating the regeneration of thymocytes and the development of thymic lymphoma following TBI.”
Reviewer 2 Report
Comments and Suggestions for Authors
The paper posits a mechanism for thymic lymphomagenesis caused by FX therapy, citing a lack of T cell precursors from bone marrow and thymus autonomy as essential components. It suggests that p53-mediated cell death in bone marrow-derived progenitors is required for thymic lymphoma formation, providing clues to prospective treatment targets. However, more elaboration of experimental methodologies, clarification of conflicting data, and the incorporation of more references are required for a thorough comprehension and scientific rigor.
Section 1:
1. While the experimental model of radiation-induced thymic lymphoma is introduced, providing additional context regarding its relevance to the broader area of cancer research may boost understanding of its significance..
2. Although the section briefly discusses the experimental technique in which mice were exposed to fractionated total-body irradiation, greater information about the experimental setting, such as dosages, timing, and specific procedures, would improve the description's comprehensiveness.
Section 2
1. The section might be better organized and structured to improve readability and flow. It contains a wide spectrum of genetic mutations and results, which might be organized and presented in a more structured format to aid comprehension.
2. Some of the findings mentioned, such as those from whole-exome sequencing, could be supplemented with brief explanations of the methodologies used, such as sample preparation, sequencing techniques, and data analysis methods, to give readers a better understanding of the research approaches used.
Section 3:
1. The section briefly mentions conflicting results regarding the methylation state of certain tumor suppressor genes, such as Pten, without providing a thorough analysis or resolution of these discrepancies. A more in-depth discussion of conflicting findings would strengthen the scientific argument.
Section 4+5
1. To improve the scientific quality of the section, the authors could broaden the discussion to include potential mechanisms underlying ROS generation in thymocytes after FX treatment, as well as the molecular pathways involved in the transformation of pre-lymphoma cells into thymic lymphoma cells.
2. Including more references to support crucial points, as well as a more extensive discussion of the findings' broader implications, would increase the section's scientific depth.
Section 6+7
1. While the section provides a thorough analysis of the experimental findings, it could benefit from clearer delineation of the methodologies used in the studies. Providing more specific details about the experimental procedures and analyses conducted would enhance the transparency and reproducibility of the research.
2. Additional references to support crucial points, as well as a more in-depth explanation of the findings' broader implications, would increase the section's scientific depth.
Sections 8-9
1. Further Elaboration: While the section provides a comprehensive overview of the proposed mechanism, further elaboration on certain aspects could enhance clarity and depth. For example, a more detailed explanation of the cellular and molecular changes involved in differentiation arrest and preneoplastic transformation could provide a clearer understanding for readers.
2. Figure Integration: The section makes reference to Figure 1, which appears to be a schematic representation of the postulated cellular activities. Integrating this graphic into the text, or offering a brief summary of its contents, would help readers visualize the proposed mechanism and improve comprehension.
3. Addressing Counterarguments: While the section presents a compelling argument for the proposed mechanism, addressing potential counterarguments or alternative explanations could strengthen the overall discussion. This could involve discussing any conflicting evidence or alternative hypotheses in the literature and explaining why the proposed mechanism is more plausible.
4. Future Directions: The section briefly mentions future directions in the summary, however commenting on prospective research routes emerging from the proposed mechanism could deepen the topic. This could involve addressing potential experimental ways to further study the postulated mechanism or looking into the translational implications for cancer treatment.
Author Response
Reviewer#2
The paper posits a mechanism for thymic lymphomagenesis caused by FX therapy, citing a lack of T cell precursors from bone marrow and thymus autonomy as essential components. It suggests that p53-mediated cell death in bone marrow-derived progenitors is required for thymic lymphoma formation, providing clues to prospective treatment targets. However, more elaboration of experimental methodologies, clarification of conflicting data, and the incorporation of more references are required for a thorough comprehension and scientific rigor.
We would like to respectfully emphasize that this review paper is not intended to discuss mechanisms or prevention of carcinogenic effects of “fractionated radiotherapy (FX therapy). Rather. we are concentrating on the mechanism of radiation carcinogenesis caused by “autonomous and non-autonomous cellular mechanism” using a classical experimental model of thymic lymphomas induced by fractionated total-body irradiation. This model was initiated by Henry Kaplan during early1950s and was described in many reference articles cited in this paper.
Section 1:
1. While the experimental model of radiation-induced thymic lymphoma is introduced, providing additional context regarding its relevance to the broader area of cancer research may boost understanding of its significance.
We thank Reviewer#2 for this comment. We have included the following paragraph in the revised manuscript to suggest the potential use of this autochthonous mouse model to study human T-ALL/T-LBL.
“Collectively, more than 70 years of research on mouse models of radiation-induced thymic lymphomas reveal cell-autonomous and non-cell-autonomous mechanisms underlying cancer initiation and progression. This knowledge provides a strong foundation that we can build upon to deeply understand cellular and molecular changes during the latency period between radiation exposure and the emergence of fully transformed cancer cells. The findings summarized in this review article may also explain the peculiar nature of the radiation dose-response curves of thymic lymphomas reported earlier for BALB/c and C57BL/6 strains that manifest a threshold around as high as 2 Gy TBI. This threshold dose may implicate that this level of radiation damage is required to reduce the number of pre-T cells in the bone marrow to cause oncogenic transformation of the surviving pro-T cells within the thymus. Furthermore, the mutational landscape of murine radiation-induced thymic lymphomas recapitulates genetic alterations in human NOTCH1 and IKZF1 frequently observed in acute lymphoblastic leukemia patients56,57. These findings suggest the potential use of this autochthonous mouse model to investigate the biology and novel treatment of human T-ALL/T-LBL.”
- Although the section briefly discusses the experimental technique in which mice were exposed to fractionated total-body irradiation, greater information about the experimental setting, such as dosages, timing, and specific procedures, would improve the description's comprehensiveness.
The experimental technique for inducing thymic lymphomas in mice has been described in our manuscript as the following:
“Experimentally, exposure of 4-week-old C57 black mice (equivalent to ~12-year-old humans) to fractionated total-body X- or γ-irradiation (FX) induces lymphomas arising from immature thymocytes 11-17. This model was first described by Dr. Henry Kaplan and colleagues in the 1950s, and their findings have been reproduced in many laboratories around the world since then. For example, the results from our laboratories show that exposure of 4-week-old C57BL/6 mice to a single fraction of 1.8 Gy total-body irradiation (TBI) per week for 4 consecutive weeks (1.8 Gy x 4) causes more than 90% of mice to develop thymic lymphomas within about 8 months after irradiation18-21. Lymphoma cells develop primarily in the thymus and express T-cell markers such as CD3, CD4, and/or CD8. In a subset of mice, lymphoma cells become leukemic and disseminate to other organs including the liver, the spleen, the kidney, and/or the bone marrow18,20. The sensitivity of mice to radiation-induced thymic lymphomas varies significantly among different mouse strains. C57BL/6 (B6) and its closely related B10 strain11-17,22-31 as well as BALB/c32, RFM33 and Swiss Albino34 strains are known to be susceptible to thymic lymphomas induced by radiation, whereas C3H35,36, STS36-39 and MSM39-42 strains are resistant. B6C3F1 mice which are F1 hybrids of susceptible B6 and resistant C3H strains are known to be susceptible43,44. Because of the robustness of generating radiation-induced blood cancers in wild-type mice, this mouse model of radiation-induced thymic lymphoma has been extensively used to study the mechanisms of radiation-induced carcinogenesis in hematopoietic cells over the past 70 years.”
Section 2
1. The section might be better organized and structured to improve readability and flow. It contains a wide spectrum of genetic mutations and results, which might be organized and presented in a more structured format to aid comprehension.
We thank Reviewer#2 for this comment. We have published two independent studies to examine the mutational landscape of murine radiation-induced thymic lymphoma (PMID: 34035082 and PMID: 37558703). In these papers, we included Figures and Tables that describe recurrent somatic mutations and copy number alterations developed in the tumors. These papers are cited in this review article for readers who are interested in knowing more about the details.
2. Some of the findings mentioned, such as those from whole-exome sequencing, could be supplemented with brief explanations of the methodologies used, such as sample preparation, sequencing techniques, and data analysis methods, to give readers a better understanding of the research approaches used.
We thank Reviewer#2 for this suggestion. We have revised our paragraph in the revised manuscript to include additional details as the following:
“To examine the mutational landscape of murine radiation-induced thymic lymphomas, we conducted whole-exome sequencing (WES) to determine somatic mutations and copy number alternations in thymic lymphomas developed in in wild-type (C57BL/6J) mice, KrasLA1 mice containing a somatic KrasG12D mutation and p53+/- mice that lose one copy of the tumor suppressor p53 in all somatic cells20. Paired normal tissues from the same mice were included as a control for the differences in germline variants. The code for replicating the figures and statistical analyses pertaining to WES results in this study are available through a public source code repository (https://gitlab.oit.duke.edu/dcibioinformatics/pubs/kirsch-lee_lymphoma). Our findings from WES demonstrate that the frequency of genetic alterations in the Notch signaling pathway is influenced by the presence of functional p5320. In lymphomas that retained wild-type (WT) p53, approximately 83% of these tumors harbored mutations in Notch1 and/or Ikzf1, a negative regulator of Notch1 signaling. In contrast to p53 WT lymphomas, mutations in Notch1 or Ikzf1 only occurred in around 38% of p53 deficient lymphomas. Nonsynonymous mutations in Notch1 were exclusively found in exon 27 which encodes the heterodimerization domain or HD (all missense mutations) and exon 34 which encodes the proline, glutamic acid, serine, threonine-rich or PEST domain (missense, stop-gain and/or frameshift mutations). Also, nonsynonymous mutations in Ikzf1 were exclusively found in exons 5/6 that encode the DNA binding domain (all missense mutations) and exon 9 (stop-gain or frameshift mutations). These mutations recapitulate hotspot mutations in human NOTCH1 and IKZF1 observed in acute lymphoblastic leukemia patients56,57. Collectively, genetic analyses of murine radiation-induced thymic lymphomas reveal a high frequency of somatic mutations and copy number variations in critical regulators of the Notch signaling. which also occurs frequently in human precursor T-cell lymphoblastic lymphoma (T-LBL) and T-cell lymphoblastic leukemia (T-ALL).”
Section 3:
1. The section briefly mentions conflicting results regarding the methylation state of certain tumor suppressor genes, such as Pten, without providing a thorough analysis or resolution of these discrepancies. A more in-depth discussion of conflicting findings would strengthen the scientific argument.
We thank Reviewer#2 for this comment. We have revised our manuscript to discuss the potential impact of mouse strains on the choice of methylations vs. genetic alterations on the tumor suppressor Pten during radiation-induced thymic lymphomagenesis in mice as the following.
“Malumbres et al. reported that inactivation of the cyclin-dependent kinase inhibitor p15INK4b due to LOH, as well as de novo methylation with independence of p16INK4a alterations, were observed among murine radiation-induced thymic lymphomas60,61. Some years later, Santos et al. also observed that Cd95 and Pten, two genes mapped at the region in chromosome 19, which were known to contain thymic lymphoma suppressors (Thymic Lymphoma Suppressor Region 8, TLSR8), were inactivated in a vast majority of these tumors (85.3% for Cd95 and 61.8% for Pten)62. These findings and the lack of mutations in the coding sequences of these genes suggest a possible regional epigenetic inactivation mechanism on mouse chromosome 19 operating during the development of radiation-induced thymic lymphomas. Song et al. also reported clear evidence for the methylation state of the promoter region of the p16 tumor suppressor gene among thymic lymphomas induced by FX treatment63. They identified 23 CpG sites of the CpG islands in the p16 promoter region and found that the methylation percentages of -71, -63, -239, -29, -38, -40, -23, 46 CpG sites were significantly higher in radiation-induced thymic lymphoma tissue than those in matched non-irradiated thymus tissue samples. They suggested that the methylation of these CpG sites in the p16 promoter may reduce its expression in the thymic lymphomas induced by radiation. On the other hand, Yamaguchi et al. reported no evidence for methylation-associated silencing of Pten. Instead, their results showed complex structural abnormalities comprised of missense and nonsense mutations, 1- and 3-bp insertions, and focal deletions in 8 of 23 lymphomas (35%)64. It is important to note that in the experiments conducted by Song et al, thymic lymphoma-induction susceptible male BALB/c strain was used63, while in the paper Yamaguchi et al used female B6C3F1 mice, which are F1 hybrid of TL-induction susceptible B6 and resistant C3H strains64. Therefore, these results suggest that both genetic and epigenetic modification of tumor suppressor genes can cooperate during the development of thymic lymphomas induced by FX treatment and reveal the potential impact of mouse strains on the mechanisms of gene silencing during radiation-induced thymic lymphomagenesis.”
Section 4+5
1. To improve the scientific quality of the section, the authors could broaden the discussion to include potential mechanisms underlying ROS generation in thymocytes after FX treatment, as well as the molecular pathways involved in the transformation of pre-lymphoma cells into thymic lymphoma cells.
- Including more references to support crucial points, as well as a more extensive discussion of the findings' broader implications, would increase the section's scientific depth.
We thank Reviewer#2 for this suggestion. Although the underlying mechanisms remain incompletely understood, we have revised our manuscript to include a paper showing that hydrogen protects mice from radiation-induced thymic lymphoma and the phenotype is associated with a decrease in ROS production. From the revised manuscript:
“In 2013, Tsuji et al. published the results that examine the dynamic changes in the cell population that occurred within the thymuses of B6 female mice at various time points after FX treatment52. Their analysis included the course of the dynamic cytogenetic as well as genomic changes of the thymocyte populations, and clonal analysis of pre-lymphoma and established lymphoma cells using TCRβ gene rearrangements. In this study, they noted that reactive oxygen species (ROS) were generated in descendants of irradiated thymocytes during recovery from radiation-induced thymic atrophy. Concomitantly, these regenerating thymus cells manifested DNA lesions as revealed by the appearance of γ-H2AX foci, chromosomal instability, and aneuploidy with trisomy 15. They also observed bystander effects on the induction of chromosome aberrations in co-cultured ROS-sensitive, XRCC4-/-, OGG1-/- and Mth1-/- mutant cells. The disappearance of these bystander effects by superoxide dismutase and catalase, suggests the role of ROS generated from post-irradiation thymocytes. Trisomy 15 and aberrant karyotypes were also observed in high frequency among these thymus cells. The emergence of thymic lymphomas from the thymocyte population containing abnormal cell clones supports that clones with trisomy 15 and altered karyotypes were pre-lymphoma cells with the potential to develop into thymic lymphomas. It was noted that alterations of the oncogene Notch1 were observed after the pre-lymphoma cells were established. In addition, Kominami and colleagues reported that Mtf-1 (metal responsive transcription factor-1) allelic genes, which are involved in the regulation of the response of cells to heavy metals and cellular ROS, play an important role in regulating the sensitivity of mice to radiation-induced thymic lymphomas39-42. Their results showed that a high level of ROS was observed in large thymocytes that survived the exposure to radiation in the thymic lymphoma-susceptible BALB/c strain, but not in thymocytes of thymic lymphoma-resistant MSM strain. Notably, one study by Zhao et al described the protection effect of hydrogen on radiation-induced thymic lymphoma in BALB/c mice. They observed that pretreatment of mice with hydrogen before TBI decreased ROS levels induced by irradiation65. Together, these findings reveal an important contribution of chronic ROS generation by regenerating thymus cells themselves during lymphomagenesis following FX treatment.”
Section 6+7
1. While the section provides a thorough analysis of the experimental findings, it could benefit from clearer delineation of the methodologies used in the studies. Providing more specific details about the experimental procedures and analyses conducted would enhance the transparency and reproducibility of the research.
- Additional references to support crucial points, as well as a more in-depth explanation of the findings' broader implications, would increase the section's scientific depth.
We consider that the methodologies used in these experiments are sufficiently given in the text and more detail in the reference papers cited in this section, except for the origin of B10.Thy 1 congenic strain mice were used in our experiments. Thus, we have revised our manuscript to include additional information.
“During the late 1970s, Sado and his collaborators at the National Institute of Radiological Sciences (NIRS) in Japan initiated their attempts to unravel the mechanism of FX-induced thymic lymphomas. They utilized B10. Thy 1 congenic pairs of mice that enabled transplantation of the thymus as a whole organ free thymocytes, or bone marrow cells, from normal or FX-treated donors, into recipients that were variously treated depending on the purpose of each experiment, and then followed the fate of donor- or host-derived cells by the use of Thy 1 alloantigen markers, Thy 1.1 or Thy 1.2.27-31. B10.Thy 1.1 congenic strain of mice used by Sado and his collaborators was bred in their laboratories before they started their thymus grafting and bone marrow transplantation experiments27 and, therefore, they are original strain combinations initiated by this group. In the first experiment of this series, Muto et al transplanted thymuses from unirradiated 7-day old donors (Thy1.1) into thymectomized recipients (Thy1.2) that were pre-treated with FX-irradiation, and the developed tumors were typed by the use of Thy 1 alloantigen markers27. To assess the origin of tumor-initiating cells, 28 out of these 37 lymphomas were typed individually and they found that 21 out of 28 thymic lymphomas (75%) originated from thymocytes of the unirradiated thymus grafts, whereas 5 tumors (17.5%) derived from cells of the irradiated hosts. Remarkably, lymphoma cells originating from unirradiated thymus grafts exhibited extensive chromosome abnormalities, which were mainly numerical changes of many chromosomes and polyploidizations, even though they were not exposed to radiation at all.”
Sections 8-9
1. Further Elaboration: While the section provides a comprehensive overview of the proposed mechanism, further elaboration on certain aspects could enhance clarity and depth. For example, a more detailed explanation of the cellular and molecular changes involved in differentiation arrest and preneoplastic transformation could provide a clearer understanding for readers.
We have added the following paragraph in the revised manuscript to extend the discussion about potential underlying mechanisms:
“When the normal cellular differentiation pathway is disrupted acutely post-TBI due to a deficit or reduction of the recruitment of T cell precursors from the bone marrow, immature thymocytes differentiating into the next stage could be arrested or frozen at a specific stage of the differentiation pathway. The differentiation arrest78-80 or maturation block81 of those immature thymocytes could be associated with the activation of aberrant expression of ‘emergency“ genes that contribute to lymphoma formation. A similar phenomenon was described by Martin et al showing that the replacement of thymus-resident precursors with bone-marrow-derived progenitors is regulated by natural cell competition between ‘young’ bone-marrow-derived and ‘old’ thymus-resident progenitors82. The authors described that thymus autonomy, a phenomenon where the thymus can produce T cells in the absence of competition from “young” bone marrow-derived lymphoid progenitors, links to a high risk for malignant transformation. Another paper from the same group showed that thymus autonomy relies on double-negative 3 early (DN3e) thymocytes that acquire stem-cell-like properties and preserve the hallmarks of thymopoiesis at the single-cell level. Notably, the authors observed a new cell population emerged in thymus autonomy, which probably serves as tumor-initiating cells because these cells express an aberrant Notch1 target gene signature83.
Figure Integration: The section makes reference to Figure 1, which appears to be a schematic representation of the postulated cellular activities. Integrating this graphic into the text, or offering a brief summary of its contents, would help readers visualize the proposed mechanism and improve comprehension.
We have extended the legends of Figure 1 in the revised manuscript as the following:
“Figure 1. Schematic representation of the possible cellular events that lead to the development of thymic lymphomas after FX treatment. In this proposed model, FX treatment induces p53-mediated cell death of bone marrow-derived hematopoietic stem/progenitor cells that impairs the supply of T cell precursors (pre-T cells) from the bone marrow to the damaged thymus. Under this condition, the progeny of the residual intrathymic T cell precursors (pro-T cells) that have survived repeated radiation exposure likely undergo differentiation or maturation arrest and thymus autonomy, a phenomenon in which resident thymocytes attempt to sustain T-cell production, but it is also permissive for preneoplastic transformation. These pre-malignant cells eventually progress to fully malignant lymphoma cells under the influence of the altered thymic microenvironment.”
Addressing Counterarguments: While the section presents a compelling argument for the proposed mechanism, addressing potential counterarguments or alternative explanations could strengthen the overall discussion. This could involve discussing any conflicting evidence or alternative hypotheses in the literature and explaining why the proposed mechanism is more plausible.
We have added the following paragraph in the revised manuscript to consider alternative mechanisms:
“Several papers published by Boniver and his collaborators show the prevention of thymic lymphomas of FX-treated C57BL mice by injection with tumor necrosis factor-α (TNF-α) after irradiation91-93. These authors started from the hypothesis that any agents able to restore the thymus after leukemogenic irradiation would exert the same effects as a BM graft. They studied TNF-α as one such possible agent because it was known to modulate the functions of the thymic epithelium and thymocyte subpopulation. Their results indicated that multiple injections of TNF-α restored thymocyte subpopulations and thymic epithelium after irradiation. These changes were associated with a decrease in PLCs and the development of radiation-induced thymic lymphomas. Moreover, BM grafts significantly stimulated intrathymic production of TNF-α messenger RNA, and anti-TNF-α antibodies partially inhibited the anti-lymphoma effects of BM grafts FX-treated mice. More recently, our group reported that BM cells from mice that have impaired tumor immunosurveillance, by deleting TNF-α, interferon γ or perforin-1, remained sufficient to suppress the formation of radiation-induced thymic lymphoma19. Together, these findings suggest that TNF-α treatment and transplantation of unirradiated BM cells may function through independent mechanisms to suppress the development of radiation-induced thymic lymphoma in mice.”
- Future Directions: The section briefly mentions future directions in the summary, however commenting on prospective research routes emerging from the proposed mechanism could deepen the topic. This could involve addressing potential experimental ways to further study the postulated mechanism or looking into the translational implications for cancer treatment.
We revised this paragraph in the revised manuscript as the following:
“Collectively, more than 70 years of research on mouse models of radiation-induced thymic lymphomas reveal cell-autonomous and non-cell-autonomous mechanisms underlying cancer initiation and progression. This knowledge provides a strong foundation that we can build upon to deeply understand cellular and molecular changes during the latency period between radiation exposure and the emergence of fully transformed cancer cells. The findings summarized in this review article may also explain the peculiar nature of the radiation dose-response curves of thymic lymphomas reported earlier for BALB/c and C57BL/6 strains that manifest a threshold around as high as 2 Gy TBI. This threshold dose may implicate that this level of radiation damage is required to reduce the number of pre-T cells in the bone marrow to cause oncogenic transformation of the surviving pro-T cells within the thymus. Furthermore, the mutational landscape of murine radiation-induced thymic lymphomas recapitulates genetic alterations in human NOTCH1 and IKZF1 frequently observed in acute lymphoblastic leukemia patients56,57. These findings suggest the potential use of this autochthonous mouse model to investigate the biology and novel treatment of human T-ALL/T-LBL.”